# CLTS-Net: A More Accurate and Universal Method for the Long-Term Prediction of Significant Wave Height

**Shuang Li** [1], **Peng Hao** [1], **Chengcheng Yu** [1,*] **and Gengkun Wu** [2]

1   Ocean College, Zhejiang University, Zhoushan 316021, China; lshuang@zju.edu.cn (S.L.); haopeng@zju.edu.cn (P.H.)
2   College of Computer Science and Engineering, Shandong University of Science and Technology, Qingdao 266590, China; wugengkun@sdust.edu.cn
*   Correspondence: chengchengyu@zju.edu.cn

**Abstract:** Significant wave height (SWH) prediction plays an important role in marine engineering areas such as fishery, exploration, power generation, and ocean transportation. For long-term forecasting of a specific location, classical numerical model wave height forecasting methods often require detailed climatic data and incur considerable calculation costs, which are often impractical in emergencies. In addition, how to capture and use the dynamic correlation between multiple variables is also a major research challenge for multivariate SWH prediction. To explore a new method for predicting SWH, this paper proposes a deep neural network model for multivariate time series SWH prediction—namely, CLTS-Net. In this study, the sea surface wind and wave height in the ERA5 dataset of the relevant points P1, P2, and P3 from 2011 to 2018 were used as input information to train the model and evaluate the model's SWH prediction performance. The results show that the correlation coefficients (R) of CLTS-Net are 0.99 and 0.99, respectively, in the 24 h and 48 h SWH forecasts at point P1 along the coast. Compared with the current mainstream artificial intelligence-based SWH solutions, it is much higher than ANN (0.79, 0.70), RNN (0.82, 0.83), LSTM (0.93, 0.91), and Bi-LSTM (0.95, 0.94). Point P3 is located in the deep sea. In the 24 h and 48 h SWH forecasts, the R of CLTS-Net is 0.97 and 0.98, respectively, which are much higher than ANN (0.71, 0.72), RNN (0.85, 0.78), LSTM (0.85, 0.78), and Bi-LSTM (0.93, 0.93). Especially in the 72 h SWH forecast, when other methods have too large errors and have lost their practical application value, the R of CLTS-Net at P1, P2, and P3 can still reach 0.81, 0.71, and 0.98. The results also show that CLTS-Net can capture the short-term and long-term dependencies of data, so as to accurately predict long-term SWH, and has wide applicability in different sea areas.

**Keywords:** CLTS-Net; significant wave height prediction; deep neural networks

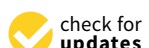



## 1. Introduction

Wave disasters are the most common marine disasters in the world. Accurate SWH prediction can effectively improve the safety of marine activities and the efficiency of marine operations, reduce the occurrence of marine accidents, and is of great significance to national security and the development of the marine economy [1–5]. Therefore, the prediction of SWH has always been a matter of special concern. Under normal circumstances, the prediction can be completed by numerical models such as WAM [6–8], WAVEWATCH [9–11], SWAN [12–14]. However, because the strong nonlinear physical process and mechanism of ocean waves are still unclear, the numerical model is still unable to obtain high accuracy to a large extent [15]. In addition, the numerical model also consumes a lot of computing resources, needs to run for a long time, and is often impractical in emergency situations such as floods or unforeseen storms that may require the transfer of personnel [16].

With the development of artificial intelligence technology, artificial neural network (ANN) models are being applied to SWH prediction [17–19]. Deo et al. [20] proposed a feed-forward network for real-time SWH prediction. Compared with traditional methods, this

method shows the advantages of versatility, flexibility, and adaptability. London et al. [21] used ANN based on existing wave datasets to predict the wave heights of six geographically separated buoy positions and found that this method has a better prediction effect in the future short-term time range. Peres et al. [22] proposed and verified an SWH prediction method based on artificial neural networks and reanalysis of wind data. This method has computational advantages. It only uses an ordinary workstation for calculation, and the calculation time is only a few hours. However, the above method can only be applied to forecasts in a relatively short period of time under normal conditions, while the forecasts under extreme conditions are not ideal. In addition, with the increase in the number of inputs and the increase in complexity, the accuracy of the ANN may drop sharply because the model cannot extract enough features [23].

Deep neural networks have received extensive attention in the marine field, and have had an extraordinary impact on solving the SWH prediction problem. The recurrent neural network (RNN) [24] model has become a more popular model in recent SWH forecasting research. Mandal et al. [25] introduced an artificial neural network RNN with a rprop update algorithm and applied it to SWH forecasting. Sadeghifar et al. [26] used RNN to predict the correlation coefficients of SWH at 3 h, 6 h, 12 h, and 24 h to be 0.96, 0.90, 0.87, and 0.73, respectively. Miky et al. [27] integrated neural network-based nonlinear autoregressive network and RNN network for SWH prediction. The experimental results show that the use of RNN for SWH prediction has better results than previous ANN methods. However, the optimization algorithm faces a significant problem during RNN training, that is, the problem of long-term dependence—due to the deepening of the network structure, the model loses the ability to learn previous information.

In response to the above problems, researchers designed a variant of RNN—namely, long short-term memory (LSTM) [28]. Compared with RNN, it is designed as a ring structure with two gated units. It can effectively solve the long-term dependence of information and avoid the disappearance or explosion of gradients, thereby significantly improving the accuracy of SWH prediction. Fan et al. [29] used the LSTM network to predict 10 sites with different environmental conditions and had better results in the 12 h, 24 h, 48 h, and 72 h SWH prediction. The results show that LSTM has a strong long-term predictive ability. Gao et al. [1] used LSTM neural network to establish a wave height prediction model at three stations in the Bohai Sea. The model uses sea surface wind and wave height as training samples to evaluate the forecasting performance of the model and perform error analysis. It is found that for SWH in the range of 3 to 5 m, the prediction accuracy of the LSTM model is significant. Zhang et al. [30] proposed the numerical long short-term memory method. This method takes the measured wave height value at the current moment and the combined wave height of the simulated nearshore wave prediction value as input, and generates the corrected numerical prediction as output. Experimental results show that this method effectively improves the SWH prediction accuracy of the Bohai Sea and Wheat Island. Raj et al. [31] developed and applied a high-precision bidirectional long-term and short-term memory (Bi-LSTM) algorithm to predict SWH and conducted overall analysis and evaluation of wave characteristics at two coastal locations in Queensland.

However, multivariate SWH forecasting still faces a major research challenge, that is, how to capture and utilize the dynamic dependencies between multiple variables. Specifically, SWH prediction models are usually a mixture of short-term and long-term dependencies. A successful SWH prediction model should capture these two dependencies to make accurate predictions. Long-term dependence considers the differences between different seasons, and short-term dependence considers wave height fluctuations caused by wind direction and wind changes in a short time. If these two dependencies are not considered, accurate SWH prediction is impossible. Therefore, solving these limitations of existing methods in SWH forecasting is the main focus of this work.

This paper proposes a deep learning model for multivariate time series SWH prediction—namely, convolutional long term time series network (CLTS-Net). As shown in Figure 1,

it uses convolutional layers to discover local dependency between multidimensional input variables; uses LSTM layers to capture complex long-term dependencies; uses a skip connection design to capture very long-term dependencies; finally, the traditional autoregressive linear model is combined with the nonlinear neural network part to make the model more robust. To better demonstrate the advantages of this method in SWH prediction, this work is compared with the currently popular ANN, RNN, LSTM, and Bi-LSTM four SWH prediction methods. The prediction results of 24 h, 48 h, and 72 h show that CLTS-Net is always better than other methods.

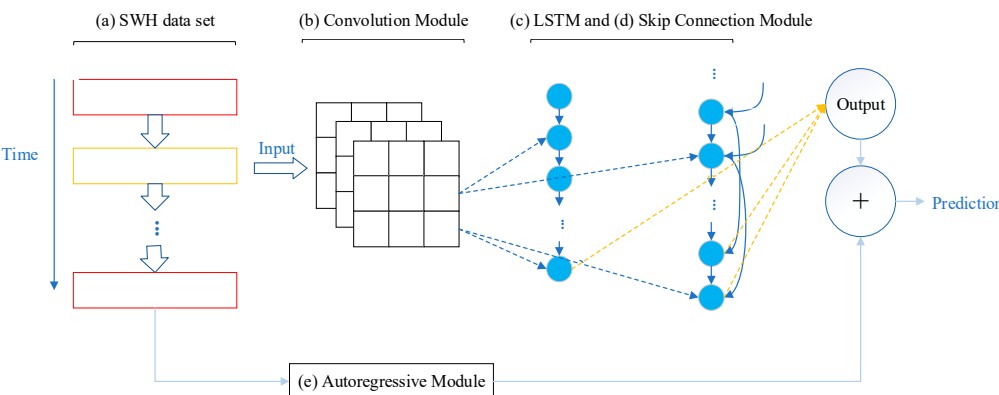

**Figure 1.** CLTS-Net overall framework.

The remainder of this paper is structured as follows: In Section 2, we describe our proposed CLTS-Net. In Section 3, the experimental design details such as the experimental dataset, metrics, and parameter settings are introduced. In Section 4, we discuss and analyze the results of SWH prediction. Finally, in Section 5, we summarize our findings.

## 2. Proposed Method

In this section, we introduce the details of the various components of the proposed CLTS-Net architecture. The overall framework of the model is shown in Figure 1.

### 2.1. Convolutional Neural Network Module

As shown in Figure 1b, the first module of CLTS-Net is composed of a convolutional neural network (CNN) without a pooling layer. Its purpose is to extract short-term patterns in the time dimension and local dependencies between variables. CNN has fewer learning parameters than standard neural networks, which contributes to trainability; in addition, CNN also shows excellent performance in successfully extracting local and translation invariant features [32]. The convolutional layer consists of two filters with a depth $d$ of 48 and a width $w$ of 3 (the width setting is the same as the number of variables). The *k-th* filter sweeps the input time series matrix $X$ and produces the corresponding calculation results. The calculation formula is as follows:

$$h_k = RELU(W_k * X + b_k) \tag{1}$$

where $*$ denotes the convolution operation, and the output $h_k$ would be a vector; the $RELU$ function is $RELU(x) = max(0, x)$, $W$ is the weight matrix, and $b_k$ is the bias.

### 2.2. Long Short-Term Memory Module

The output of the convolutional layer is simultaneously input to the LSTM module in Figure 1c and the jump connection module in Figure 1d. As shown in Figure 2, LSTM uses two gates to control the content of the cell state $c$: one is the forget gate, which determines how much the cell state $c_{t-1}$ from the previous moment is retained to the current moment $c_t$; the other is the input gate, which determines how much of the input $x_t$ of the network at the current moment is saved in the unit state $c_t$. The LSTM module uses an output

gate to control how much of the unit state $c_t$ is input to the current output value $h_t$ of the LSTM module.

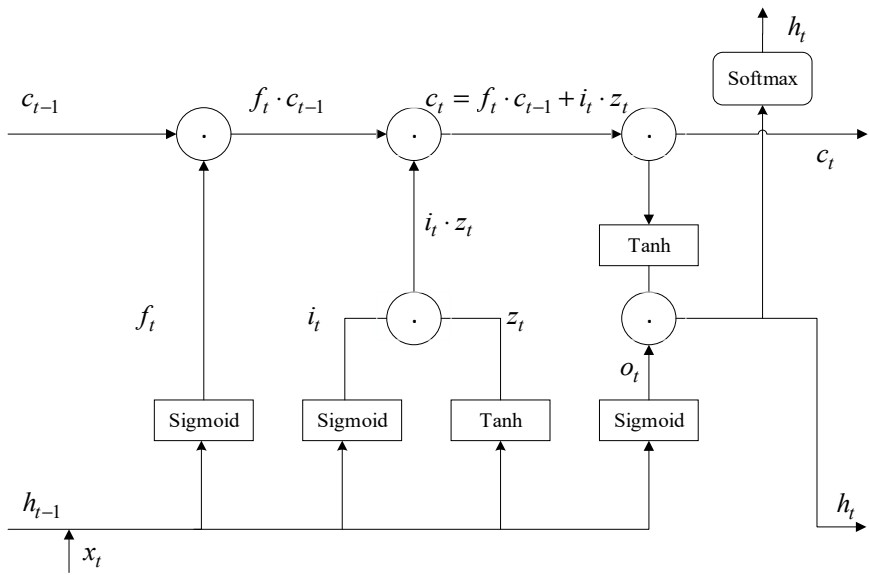

**Figure 2.** LSTM module architecture.

This module uses the tanh function as the activation function, and the information state transfer formula of the unit at time $t$ in LSTM is as follows:

$$f_t = \sigma\left(w_f[h_{t-1}, x_t]\right) \tag{2}$$

$$i_t = \sigma(w_i[h_{t-1}, x_t]) \tag{3}$$

$$z_t = \tanh(w_z[h_{t-1}, x_t]) \tag{4}$$

$$c_t = f_t c_{t-1} + i_t \cdot z_t \tag{5}$$

$$o_t = \sigma(w_o[h_{t-1}, x_t]) \tag{6}$$

$$h_t = o_t \cdot \tanh(c_t) \tag{7}$$

where $f_t$ represents the processing formula of the forget gate, $i_t$ represents the processing formula of the input gate, $o_t$ represents the processing formula of the output gate, $w$ represents the given weight coefficient, $\sigma$ represents the sigmoid function, and $\cdot$ represents the element-wise product.

### 2.3. Skip Connection Module

The meticulous design of the LSTM module is used to memorize historical information so that the model can learn relatively long-term dependencies. However, due to the disappearance of the gradient, LSTM often fails to capture very long-term correlations in practice. Inspired by the Res-Net [33], we aimed to alleviate this problem by jumping connections. As shown in Figure 1d, we used this design to extend the time span of the information flow, thereby capturing the longer-term dependence of the data. In practical applications, we can better improve the accuracy of the model by analyzing the characteristics of the dataset and selecting an appropriate time span. We used a dense layer to combine the outputs of the LSTM and skip connection module. The output formula of this module is as follows:

$$h_t^D = W^L h_t^L + \sum_{i=0}^{m-1} W_i^S h_{t-i}^S + b_S \tag{8}$$

where $h_t^D$ is the prediction result of the neural network, $h_t^L$ is the output of the LSTM, $m$ is the number of hidden units skipped, $b_S$ is the bias, and $h_{t-m+1}^S, h_{t-m+2}^S, \ldots, h_t^S$ means jump $m$ hidden states that have been passed.

### 2.4. Autoregressive Module

In the SWH dataset, the input data are constantly changing in a nonperiodic manner, which greatly reduces the prediction accuracy of the neural network model. To solve this defect, we decomposed the final prediction of CLTS-Net into a linear part that mainly focuses on local-scale problems and a nonlinear part that contains repeated patterns. As shown in Figure 1e, we used the classic autoregressive model as the linear module. We denoted the forecasting result of the autoregressive model as $h_t^R \in \mathbb{R}$, and $b^{ar} \in \mathbb{R}$, where $q^{ar}$ is the size of the input window over the input matrix. The autoregressive model is formulated as follows:

$$h_{t,i}^R = \sum_{j=0}^{q^{ar}-1} W_j^{ar} y_{t-j,i} + b^{ar} \tag{9}$$

Overall, the CLTS-Net model contains two parts—one is a nonlinear neural network model part and a linear autoregressive part. The final result is the addition of the output of these two parts,

$$F = h_t^D + h_t^R \tag{10}$$

where $F$ denotes the CLTS-Net's final prediction at time $t$.

## 3. Evaluation

In this section, we first introduce the research area and the source of the dataset used in this paper; then, we introduce other methods and metrics to be compared in the experiment; finally, we describe the experimental environment.

### 3.1. Dataset

ERA5 is the fifth-generation ECMWF reanalysis for the global climate and weather for the past four to seven decades. As shown in Figure 3, we selected (latitude 29°~31° N, 120°~130° E) as the study area. This area is dominated by wind and waves and is greatly affected by the monsoon. The time resolution of the data is hours, and the spatial resolution is 0.5° × 0.5°. For more information, please refer to the website: https://cds.climate.copernicus.eu/cdsapp#!/dataset/reanalysis-era5-single-levels?tab=form (accessed on 16 December 2021).

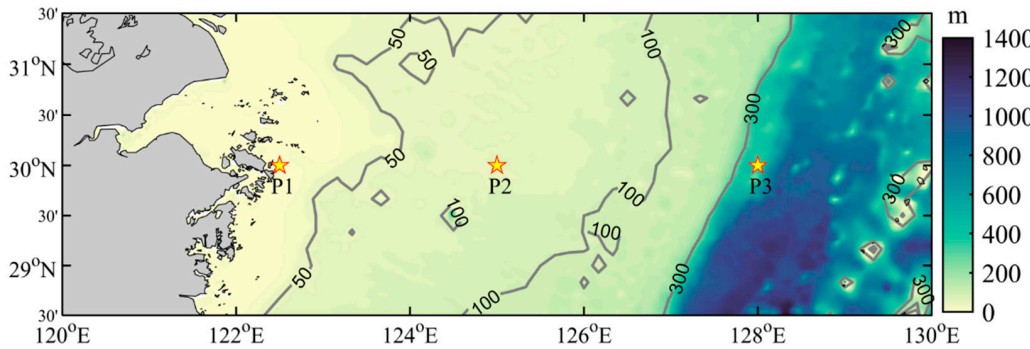

**Figure 3.** Position distribution of the three stations.

To test the stability and reliability of the model, we selected three sites (P1, P2, P3) with different water depths and different environmental conditions. For the prediction of SWH, we used the data from 2011 to 2018 to generate the corresponding training set and the last 720 h of data in 2020 as the test set. To ensure the relative independence of training and test datasets, the test data were excluded from model training.

### 3.2. Methods for Comparison

The methods in our comparative evaluation are as follows:

ANN: The network achieves the purpose of processing information by adjusting the connections between a large number of internal nodes;

RNN: A neural network that is used to process sequence data, which has certain advantages when learning the nonlinear characteristics of the sequence;

LSTM: This module can effectively solve the long-term dependence of information and avoid the disappearance or explosion of the gradient;

Bi-LSTM: This method is a combination of forward LSTM and backward LSTM;

CLTS-Net: This is the method proposed in this paper, which combines the advantages of the CNN, LSTM, and autoregressive models. It captures the short-term and long-term dependencies in the data and combines linear and nonlinear models to make robust predictions.

### 3.3. Metrics

To evaluate the performance of the model, we used the following four metrics—namely, root-mean-square error (RMSE), mean absolute error (MAE), mean absolute percent error (MAPE), and correlation coefficient (R).

$$MAE = \frac{1}{n} \sum_{i=1}^{n} |\hat{y}_i - y_i| \tag{11}$$

$$RMSE = \sqrt{\frac{1}{n} \sum_{i=1}^{n} (\hat{y}_i - y_i)^2} \tag{12}$$

$$MAPE = \frac{100\%}{n} \sum_{i=1}^{n} \left| \frac{\hat{y}_i - y_i}{y_i} \right| \tag{13}$$

$$R = \frac{\sum_{i=1}^{n} (\hat{y}_i - \overline{\hat{y}_i})(y_i - \overline{y_i})}{\sqrt{\sum_{i=1}^{n} (\hat{y}_i - \overline{\hat{y}_i})^2 \sum_{i=1}^{n} (y_i - \overline{y_i})^2}} \tag{14}$$

In the formula, $n$ is the total number of test samples, and $y_i$ and $\hat{y}_i$ are the true and predicted values, respectively. It is worth noting that the lower the RMSE, MAE, and MAPE values, the better the consistency between the measurement and the prediction, but the higher the R-value, the more accurate the prediction.

### 3.4. Experimental Details

Except for the input layer and output layer, we performed a dropout procedure after each layer, and the rate was set to 0.2. The Adam algorithm was used to optimize the model parameters.

We used a 2.60 GHz Intel Core i5-11400F processor, graphics card Nvidia GeForce RTX 3060 computing platform for experiments. The methods mentioned in the experiment were all implemented by Keras in the Python environment.

## 4. Results

We conducted multiple sets of experiments to verify the performance of each model to predict SWH and analyzed the experimental results, the discussion of which is presented in this section.

### 4.1. SWH Forecast Performance at P1

Table 1 list the experimental results of five algorithms at P1 after training, verification, and testing. The best results are shown in bold. It is worth noting that the prediction results of SWH short-term forecasts (1 h, 3 h, 6 h, and 12 h) based on deep learning methods

have reached high accuracy and are not the focus of this work. This paper only focuses on longer-term forecasts (24 h, 48 h, and 72 h).

**Table 1.** The prediction result of significant wave height at point P1.

| Metrics | Method | 24 h | 48 h | 72 h |
|---------|--------|------|------|------|
| RMSE | ANN | 0.2881 | 0.3520 | 0.4102 |
| | RNN | 0.2750 | 0.2668 | 0.3199 |
| | LSTM | 0.2182 | 0.2044 | 0.3124 |
| | Bi-LSTM | 0.1102 | 0.1413 | 0.3076 |
| | CLTS-Net | **0.0424** | **0.0465** | **0.2008** |
| MAE | ANN | 0.1864 | 0.2470 | 0.2837 |
| | RNN | 0.1821 | 0.1796 | 0.1876 |
| | LSTM | 0.1509 | 0.1309 | 0.1685 |
| | Bi-LSTM | 0.0628 | 0.0817 | 0.1486 |
| | CLTS-Net | **0.0270** | **0.0280** | **0.0706** |
| MAPE | ANN | 0.2505 | 0.3472 | 0.3773 |
| | RNN | 0.2802 | 0.2750 | 0.2631 |
| | LSTM | 0.2426 | 0.1924 | 0.2234 |
| | Bi-LSTM | 0.0927 | 0.1159 | 0.1818 |
| | CLTS-Net | **0.0469** | **0.0447** | **0.0764** |
| R | ANN | 0.7966 | 0.7010 | 0.2218 |
| | RNN | 0.8274 | 0.8357 | 0.4559 |
| | LSTM | 0.9385 | 0.9180 | 0.4772 |
| | Bi-LSTM | 0.9593 | 0.9420 | 0.5017 |
| | CLTS-Net | **0.9913** | **0.9911** | **0.8054** |

It can be seen from the results that the predictive index R of ANN in 24 h is maintained at about 0.79. When it is greater than 48 h, the prediction error of ANN is too large to apply and predict long-term SWH, while the prediction result of RNN is better than 24 h. The prediction performance of LSTM and Bi-LSTM is better within 24 h. When it exceeds 48 h, the prediction performance decreases significantly.

Overall, as the time span increases, the MAE, RMSE, and MAPE of each method show an increasing trend, and their R gradually decreases. In particular, when the prediction time increases by more than 24 h, the indicators of other algorithms are significantly reduced, and CLTS-Net can maintain good prediction results at 24 h and 48 h. Even in the 72 h forecast, R can still reach 0.80, which is still significantly better than other methods.

From the experimental results obtained in Table 1, it can be seen that when predicting the 72 h SWH, only CLTS-Net can maintain better prediction performance. The performance of the other methods is compared with the 24 h measurement index. There is a considerable decline, and it loses the value of a practical application, therefore the meaning of comparison. In order to facilitate a more intuitive display of the SWH prediction results, Figures 4 and 5 show the 24 h and 48 h SWH prediction results, respectively. In general, it can be seen that the performance of CLTS-Net is better than the other four methods. In most periods, the prediction curve of each point is almost the same as the ERA5 data curve. To better show the prediction effect of the method in this paper, we separately provide the corresponding fitting curve diagram.

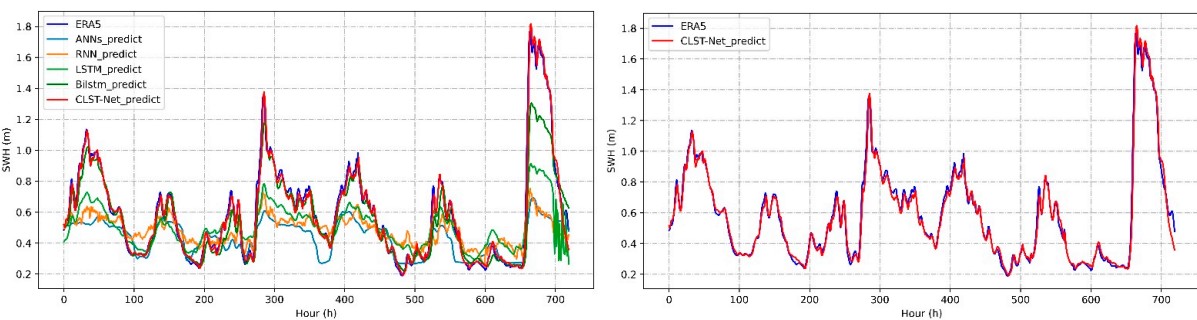

**Figure 4.** Continuous prediction at point P1 for 24 h prediction.

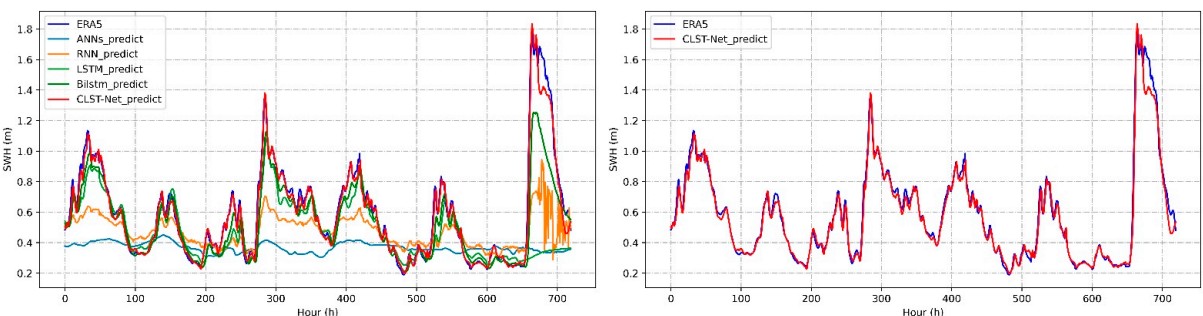

**Figure 5.** Continuous prediction at point P1 for 48 h prediction.

Figures 4 and 5, respectively, show the comparison between the observed and predicted values of CLTS-Net's continuous forecast every 24 h and 48 h at point P1. Since point P1 is located in the offshore area, the wave height is relatively low throughout the year, and the smaller the overall error is smaller. It shows that CLTS-Net is feasible to predict the long-term SWH of the 48 h time span in the offshore area.

We focused on analyzing the errors of the 24 h and 48 h prediction results at various points and troubleshooting the reasons, so that future optimization work can be better carried out.

At point P1, the wave height range of ERA5 is between 0.2 m and 2 m. The scattering point closer to the diagonal reflects the higher the accuracy of the model's prediction. It can be seen from Figures 6 and 7 that the prediction results of CLTS-Net are closer to the diagonal, while the scattered points of other methods (ANN, RNN, LSTM) are concentrated on the lower right of the diagonal, which means that the results tend to low.

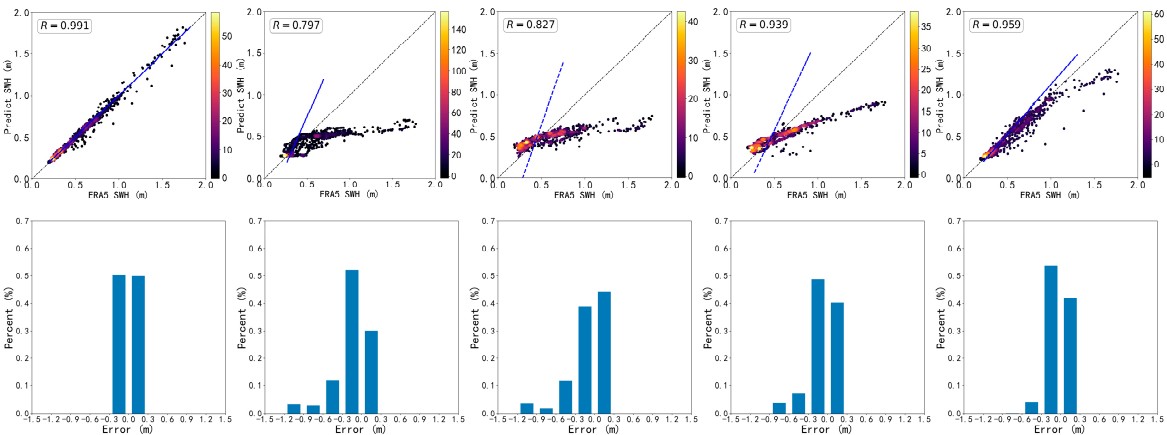

**Figure 6.** Each column represents a scatter diagram (**top**) and error range diagram (**bottom**) of a method for continuous forecasting at point P1 for 24 h forecasts. From left to right, they are CLTS-Net, ANN, RNN, LSTM, Bi-LSTM.

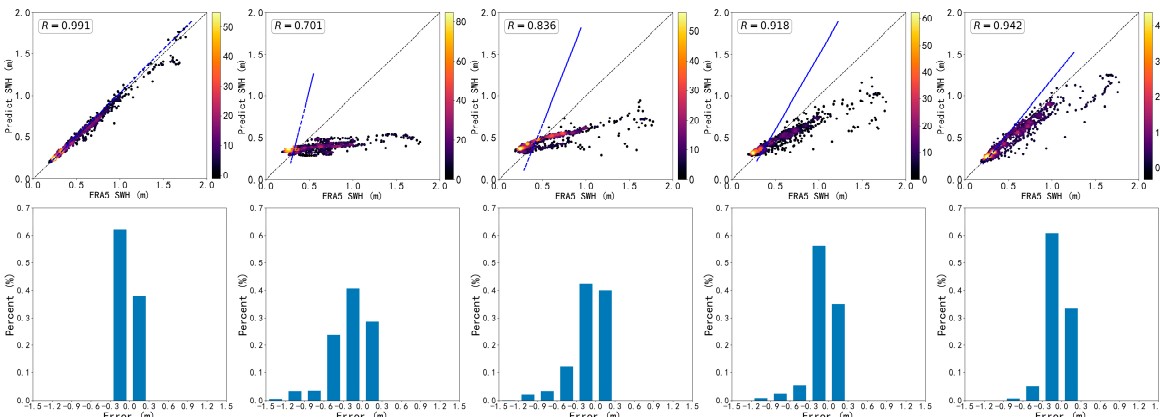

**Figure 7.** Each column represents a scatter diagram (**top**) and error range diagram (**bottom**) of a method for continuous forecasting at point P1 for 48 h forecasts. From left to right, they are CLTS-Net, ANN, RNN, LSTM, and Bi-LSTM.

When the SWH is about 0.5 m, the prediction results of ANN, RNN, and LSTM are better. When the SWH value becomes larger, the predicted value will have a larger deviation, which may be because point P1 is located in the coastal area, and the training data lack a relatively large SWH value. From the perspective of prediction results, the bidirectional LSTM structure can indeed capture more feature information and dependencies, and the prediction effect is better than LSTM, but as the SWH value increases, the predicted value will also appear low problem. The advantage of CLTS-Net at point P1 is obvious, i.e., the error is concentrated in the range of ±0.3 m, and the 24 h and 48 h correlation coefficients are as high as 99%.

### 4.2. SWH Forecast Performance at P2

Table 2 list the experimental results of five algorithms at P2 after training, verification, and testing. The best results are shown in bold.

**Table 2.** The prediction result of significant wave height at point P2.

| Metrics | Method | 24 h | 48 h | 72 h |
|---------|--------|------|------|------|
| RMSE | ANN | 0.7611 | 1.0941 | 1.1397 |
| | RNN | 0.6852 | 0.7564 | 1.1105 |
| | LSTM | 0.5001 | 0.6299 | 1.0183 |
| | Bi-LSTM | 0.4563 | 0.6117 | 0.9827 |
| | CLTS-Net | **0.1314** | **0.1824** | **0.7740** |
| MAE | ANN | 0.4589 | 0.6798 | 0.7015 |
| | RNN | 0.3162 | 0.3819 | 0.6290 |
| | LSTM | 0.2673 | 0.2972 | 0.3790 |
| | Bi-LSTM | 0.2045 | 0.3091 | 0.3801 |
| | CLTS-Net | **0.0618** | **0.0868** | **0.2444** |
| MAPE | ANN | 0.2271 | 0.2879 | 0.3038 |
| | RNN | 0.1252 | 0.1910 | 0.2564 |
| | LSTM | 0.1372 | 0.1151 | 0.1233 |
| | Bi-LSTM | 0.0823 | 0.1324 | 0.1310 |
| | CLTS-Net | **0.0276** | **0.0389** | **0.0757** |
| R | ANN | 0.7616 | 0.6814 | 0.2157 |
| | RNN | 0.8381 | 0.7214 | 0.4139 |
| | LSTM | 0.9026 | 0.8886 | 0.3993 |
| | Bi-LSTM | 0.9387 | 0.9215 | 0.4495 |
| | CLTS-Net | **0.9921** | **0.9883** | **0.7107** |

Compared with the model performance at point P1, the predictive indicators at point P2 are generally lower. The possible reason is that the SWH at point P2 is relatively large and changes relatively quickly.

Figures 8 and 9, respectively, show the comparison between the observed and predicted values of CLTS-Net's continuous forecast every 24 h and 48 h at point P2. For point P2, it is relatively far away from the land, and the wave heights are mostly 2–6 m, and there are even more than 6 m in some periods. It can be seen from the curve that, except for the method in this article, the other methods cannot accurately predict the sudden increase in wave height, and the prediction result is always far smaller than the real value. However, the method proposed in this paper fits better when the wave height is low and also has a small error even when the wave height is greater than 6 m, indicating that CLTS-Net is feasible for 48 h long-term SWH prediction.

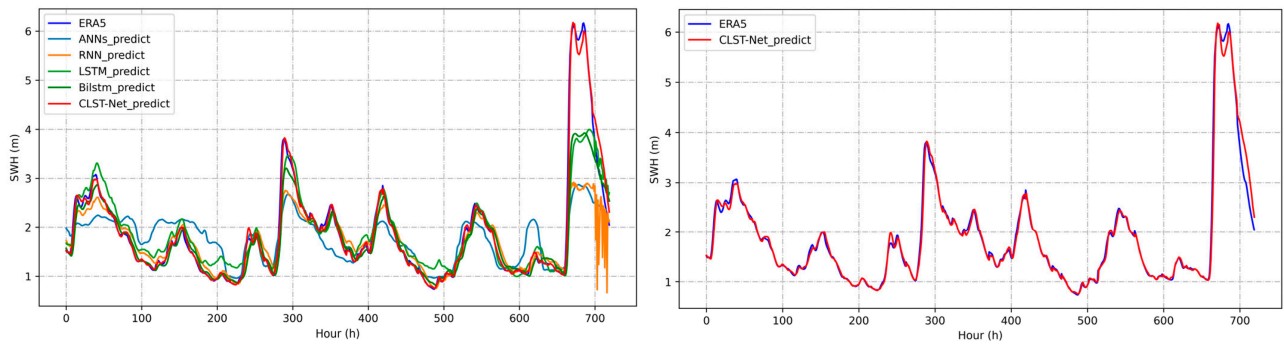

**Figure 8.** Continuous prediction at point P2 for 24 h prediction.

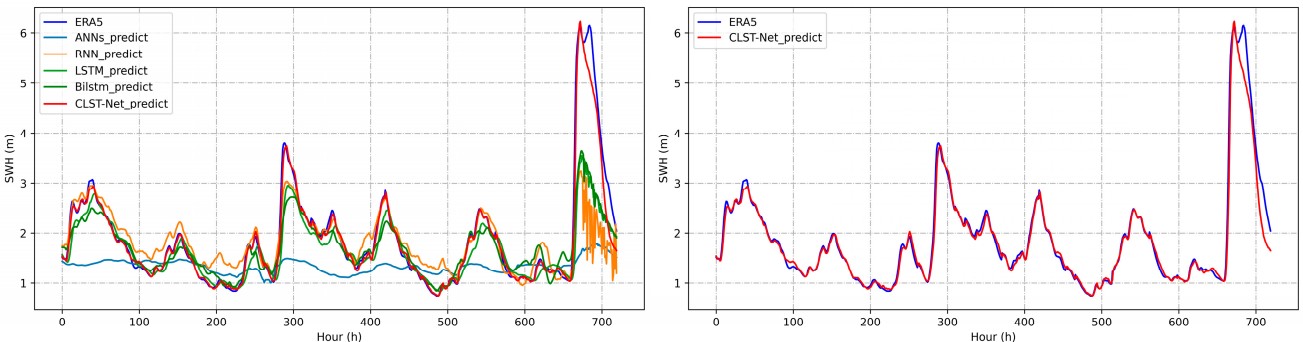

**Figure 9.** Continuous prediction at point P2 for 48 h prediction.

At point P2, the wave height range of ERA5 is between 0.5 m and 6.5 m, with a relatively large range of change and a sharper trend. Compared with points P1 and P3, the overall error is larger. As shown in Figures 10 and 11, CLTS-Net has some prediction results that are too large in the 24 h prediction, but the overall gap is not large, and the error is mostly about ±0.5 m, which is within an acceptable range; the prediction result of ANN is not ideal whether it is the prediction result of 24 h or 48 h; from RNN to LSTM to Bi-LSTM, the error is gradually reduced, and the R-value of the prediction is gradually improved, but compared with CLTS-Net, these three methods still have a large number of points with large errors, and even some points deviate by more than 3 m, which is an unacceptable prediction result.

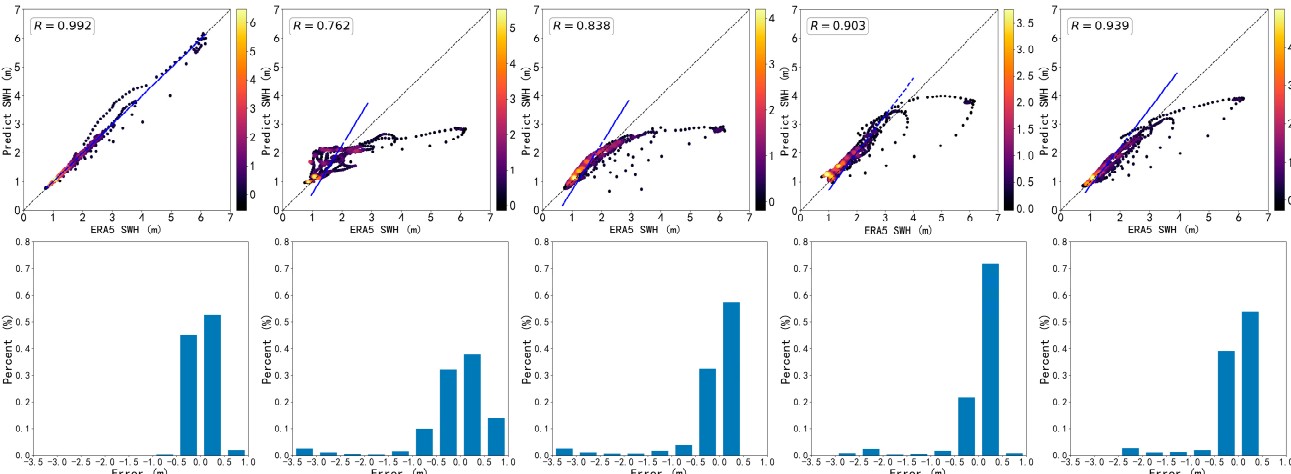

**Figure 10.** Each column represents a scatter diagram (**top**) and error range diagram (**bottom**) of a method for continuous forecasting at point P2 for 24 h forecasts. From left to right, they are CLTS-Net, ANN, RNN, LSTM, and Bi-LSTM.

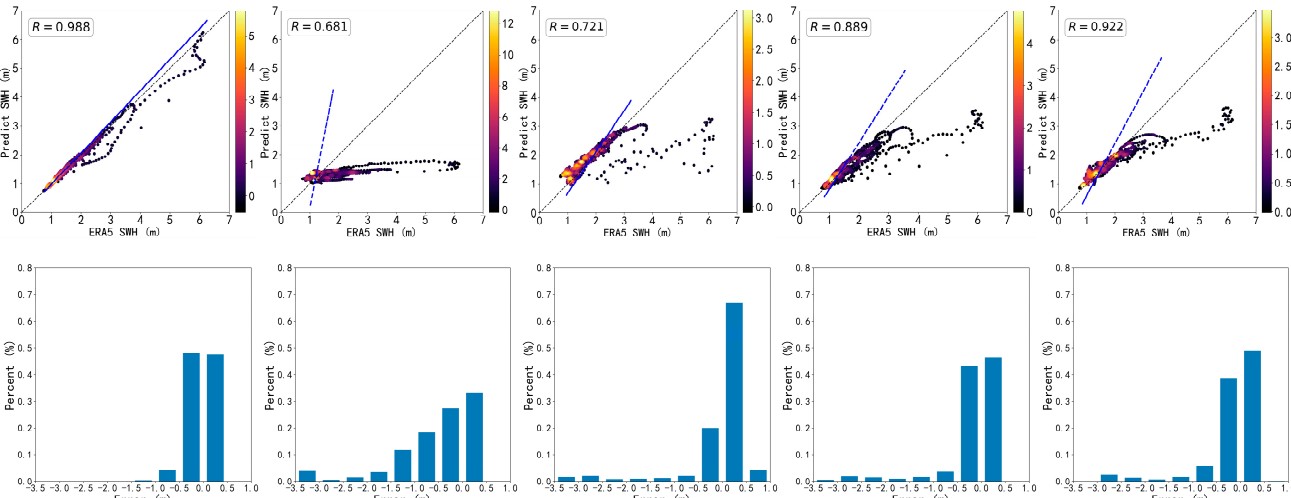

**Figure 11.** Each column represents a scatter diagram (**top**) and error range diagram (**bottom**) of a method for continuous forecasting at point P2 for 48 h forecasts. From left to right, they are CLTS-Net, ANN, RNN, LSTM, and Bi-LSTM.

### 4.3. SWH Forecast Performance at P3

Table 3 list the experimental results of five algorithms at P3 after training, verification, and testing. The best results are shown in bold.

It is worth noting that, compared with points P1 and P2, the 72 h prediction correlation coefficient of point P3 can still reach 0.98. The reason may be while the model fully learns the characteristics, the data changes at this stage are relatively regular, and the data fluctuations are relatively stable. The conclusions in Tables 1–3 can illustrate that capturing short-term and long-term dependencies helps predict long-term SWH and also illustrates the wide applicability of CLTS-Net in different sea areas.

Figures 12 and 13 are the comparison of the observed and predicted values of CLTS-Net's continuous forecast every 24 h and 48 h at point P3, respectively. Point p3 is located in the deep-sea area, and the wave height changes rapidly. Still the same problem, other methods cannot accurately predict the sudden increase in wave height, and the prediction result is always smaller than the actual value.

**Table 3.** The prediction result of significant wave height at point P3.

| Metrics | Method | 24 h | 48 h | 72 h |
|---------|--------|------|------|------|
| RMSE | ANN | 0.6239 | 0.6415 | 0.7029 |
| | RNN | 0.3792 | 0.5715 | 0.5917 |
| | LSTM | 0.3009 | 0.3092 | 0.5161 |
| | Bi-LSTM | 0.2868 | 0.2681 | 0.4990 |
| | CLTS-Net | **0.1647** | **0.1271** | **0.1283** |
| MAE | ANN | 0.4676 | 0.5020 | 0.5319 |
| | RNN | 0.3053 | 0.4470 | 0.4650 |
| | LSTM | 0.2098 | 0.2049 | 0.4058 |
| | Bi-LSTM | 0.1964 | 0.1772 | 0.3649 |
| | CLTS-Net | **0.1275** | **0.0914** | **0.0900** |
| MAPE | ANN | 0.2341 | 0.2497 | 0.2745 |
| | RNN | 0.1985 | 0.2418 | 0.2663 |
| | LSTM | 0.1151 | 0.1082 | 0.2350 |
| | Bi-LSTM | 0.0999 | 0.0889 | 0.1944 |
| | CLTS-Net | **0.0760** | **0.0524** | **0.0553** |
| R | ANN | 0.7101 | 0.7234 | 0.3123 |
| | RNN | 0.8552 | 0.7850 | 0.6260 |
| | LSTM | 0.8950 | 0.9117 | 0.6649 |
| | Bi-LSTM | 0.9377 | 0.9315 | 0.6971 |
| | CLTS-Net | **0.9736** | **0.9850** | **0.9844** |

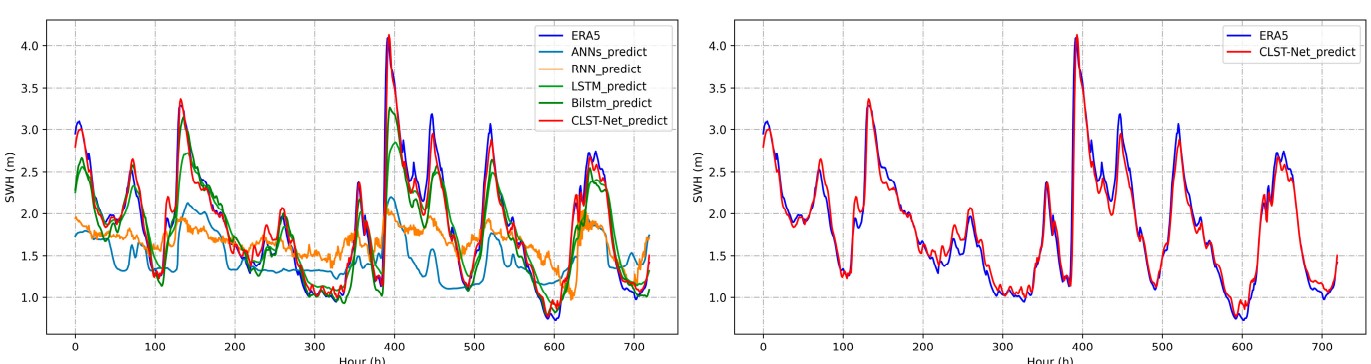

**Figure 12.** Continuous prediction at point P3 for 24 h prediction.

**Figure 13.** Continuous prediction at point P3 for 48 h prediction.

At point P3, the wave height range of ERA5 is between 0.5 m and 4.5 m, which is smaller than the wave height change range of point P2, but the changing trend is more intense. As shown in Figures 14 and 15, CLTS-Net has some small prediction results in the 24 h prediction, but the overall gap is not large; the error distribution is about ±0.5 m, which is still within an acceptable range. Regardless of whether the prediction result of ANN is 24 h or 48 h, there are still large errors, indicating that the method of ANN is not suitable for long-term SWH prediction; in addition, regardless of whether RNN forecast is at 24 h or 48 h, there are many cases in which the forecast value is low, many data errors exceed 1 m, and the accuracy and reliability methods are poor; the accuracy of Bi-LSTM is slightly improved compared with LSTM, but there are also data with large errors, and the overall prediction value is low, which is not an effective SWH solution.

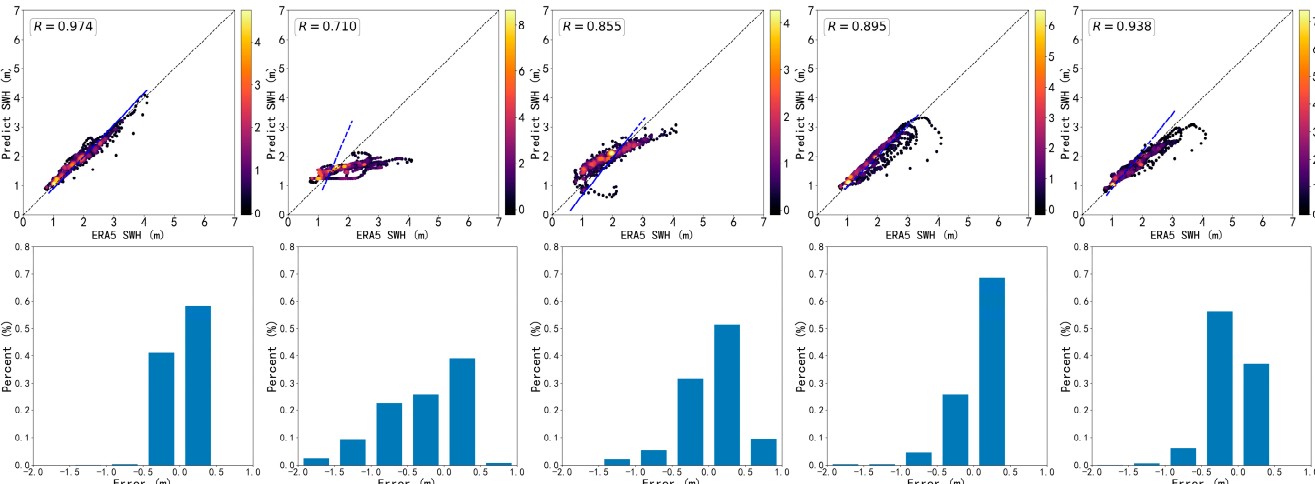

**Figure 14.** Each column represents a scatter diagram (**top**) and error range diagram (**bottom**) of a method for continuous forecasting at point P3 for 24 h forecasts. From left to right, they are CLTS-Net, ANN, RNN, LSTM, and Bi-LSTM.

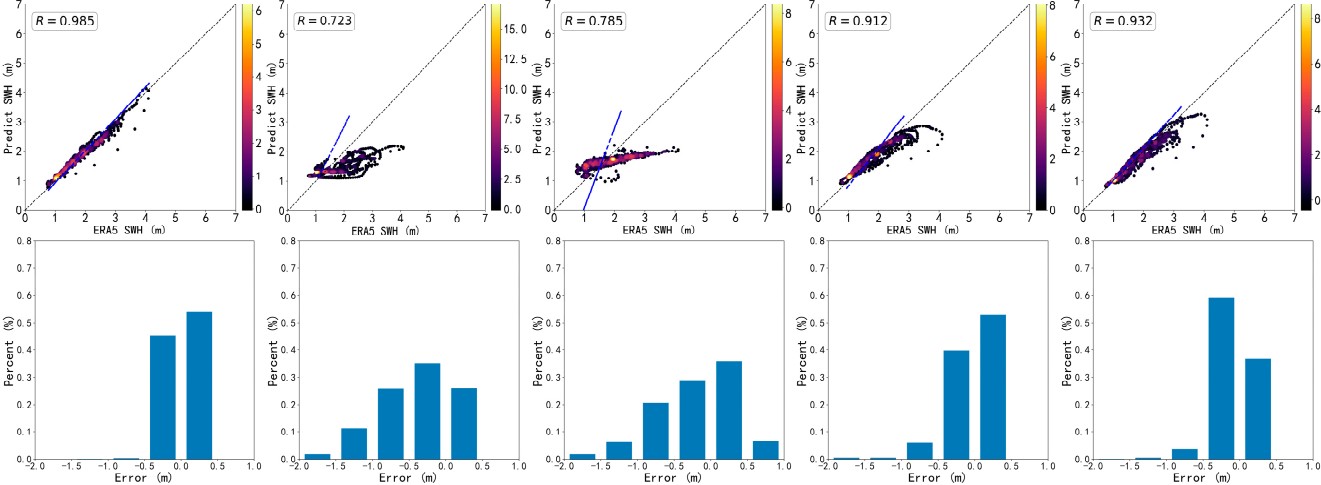

**Figure 15.** Each column represents a scatter diagram (**top**) and error range diagram (**bottom**) of a method for continuous forecasting at point P3 for 48 h forecasts. From left to right, they are CLTS-Net, ANN, RNN, LSTM, and Bi-LSTM.

## 5. Conclusions

This paper proposed a deep learning model for multivariate time series SWH prediction. First, the method used convolutional layers to discover local dependency patterns between multidimensional input variables and used LSTM layers to capture complex long-term dependencies; then, jump connections were used designed to capture very long-term

dependencies; finally, the traditional autoregressive linear model and the nonlinear neural network part were paralleled to make the model more robust.

To compare the quality of the model, we used five algorithms to predict the SWH of three different stations under different marine environmental conditions. We used the complete data of these sites from 2011 to 2018 to train the model and used four indicators to evaluate the accuracy and stability of the prediction results. The results show that the CLTS-Net algorithm can obtain more accurate results in 24 h, 48 h, and 72 h predictions.

It can be seen that the SWH prediction technology based on CLTS-Net can make full use of the important information of sea wind and significant wave height, establish a prediction model, and realize business applications. This method opens up a new field for ocean forecasting and has broad prospects for development and application. For future research, there are several promising directions to extend this work. Due to the complexity of the actual marine environment, it is quite a challenging task to extend the CLTS-Net method to all sites based on a broader dataset. The number of input features directly determines the prediction results. For this reason, more environmental factors, such as wind speed, water depth, terrain, etc., need to be considered and added to the input of CLTS-Net. This general deep learning model deserves more attention in future analysis. In addition, the current work mainly considers single-point forecasts. In the future, it will be considered to extend single-station forecasts to regional forecasts.

**Author Contributions:** Methodology, P.H.; software, P.H. and G.W.; validation, S.L.; formal analysis, S.L. and C.Y.; investigation, S.L.; resources, S.L.; data curation, P.H.; writing—original draft preparation, P.H.; writing—review and editing, S.L.; visualization, P.H.; supervision, S.L. All authors have read and agreed to the published version of the manuscript.

**Funding:** This research was funded by the National Key Research and Development Plan of China (Grant Number 2017YFA0604101), the China-Sweden (NSFC-STINT) cooperation and Exchange Project under contract (Grant Number 41911530149), and the National Natural Science Foundation of China (Grant Number 41876003 and 41830533).

**Institutional Review Board Statement:** Not applicable.

**Informed Consent Statement:** Not applicable.

**Data Availability Statement:** For more information, please refer to the website: https://cds.climate.copernicus.eu/cdsapp#!/dataset/reanalysis-era5-single-levels?tab=form (accessed on 16 December 2021).

**Acknowledgments:** We would like to acknowledge the organizations that provided the sources of the data used in this work—namely, the European Centre for Medium-Range Weather Forecasts.

**Conflicts of Interest:** The authors declare no conflict of interest.

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
