# Peer review of "CLTS-Net: A More Accurate and Universal Method for the Long-Term Prediction of Significant Wave Height"

_jmse, doi:10.3390/jmse9121464_

Round 1
Reviewer 1 Report
Congratulations on the clear and interesting article. The manuscript addresses an important topic, even though many papers have been published on this, with similar models being developed. Nevertheless, I believe the paper is publishable, since it does have novelty by presenting a new model that outperforms others, which are well established in the literature. At this point in time, the model is till much related to this site specific case. I wonder if the excellent performance is maintained in other cases. Still, this does not affect the quality of the paper, although it impacts its outreach if the methodology fails to model so well other locations. Something to explore in future works for sure. Something to improve is the notion on how important the model can be for practical applications (see my comments related to the intro and the conclusions). References also need improvement. There are few to almost none typos. The paper is good and I enjoyed reading it for sure! I believe that it has interesting practical influence for many maritime engineering and science applications for both coastal and offshore environments. Congratulations once again! Please address my minor comments below.
L33 - 37: references 1 to 3 do not address specific disasters, human, environmental and assets losses. They do serve as a point to justify the paragraph. However in the next sentence L36 to 37 "Therefore...special concern", I recommend the inclusion of 2 additional references, which do provide information on such aspects, namely with focus on losses related to the important fields of marine energy and oil & gas offshore industry: REF A doi - 10.1680/jmaen.2019.172.4.118 and REF B doi - 10.1680/jmaen.2019.20
In the text between the words and the numbers of the references the space is missing.
L43 What do you mean by emergencies? Some examples would be helpful, e.g. sudden coastal floods or offshore infrastructures that may need people to be removed due to unforeseen storms or so?
L61-62 rprop? is this a typo?
L73: what do you mean by explosion of gradients? Please elaborate on this.
L89-97: Recently many multivariate models have been published, including interesting applications to marine engineering, coastal and offshore applications etc...I remember for example the works of Jonathan Tawn, Ewins, Vanem among many others... Although not exactly through neural networks but using different prediction approaches, with copulas, asymmetric models, conditional model approach among others. These works provide numerous applications and highlight the importance of multivariate analysis as the one presented here, e.g. DOI: 10.1177/0309524X18777323 or DOI: 10.1177/0309524X18807033 or DOI: 10.1080/01621459.2021.1996379 Perhaps it would be nice to include a comment on these and some of these examples, stating how important the model you propose can benefit further maritime engineering applications.
L99 convolutional
L101 define LSTM in extended format in the first time you use it.
L120-121: why having few parameters contributes to trainability? Please elaborate on this.
Models in section 2 should have at least 1 or 2 references that explain them with a bit more detail, for the benefit of the reader.
L190: selected instead of select
L199-201: what is the number of hours used for the training data set? Wouldn't a very large amount of training data compared to testing data lead to very successfull results in the model's accuracy that are then not directly translated to other situations where this proportion more "equilibrated"?
L206 - which certain advantages? be more specific for the benefit of the reader.
L242: I am not totally fan of naming 24, 48 and 72 hours long-term predictions, as in most engineering applications and oceanography studies, long-term typically relates to SWH with large return periods, e.g. years. Anyway, you can keep it as it is but this needs to be clarified at the beginning of the intro.
Table 1 - remove the (Our) from the Table, we now the method is yours :)
L250 - correct: "decreases.in..."
L252: you often refer "our method", for the sake of the scientific writing expected at a scientific journal please use the actual method's name instead of our method. Correct this throughout the paper.
Figure 4 and 5 - P1 has been shown separated from P2 and P3. However, putting them together would help to evaluate if for example the water depth has an influence in the model's results.
Also it seems that the models, which perform worse, tend to underestimate SWH. Please add a comment about how this has practical implications for related disasters and so on.
Figure 6 - you should adopt the same order as in Table 1 to avoid confusions
L336 how was the "acceptable range" defined? If it is a similar acceptable range as in other works of the speciality, then citations need to be included here.
Perhaps I am seeing it wrongly, but it seems that larger deviations for the novel model occur for larger water depths P3. It would be nice to apply this model to other locations, to see if this still occurs, because, often in coastal applications having good predictions at P1 is more important than in P3. On the other hand, for offshore applications, good predictions at P2 and P3 are more important than in P1. Anyway, the model does seem to clearly outperform the remaining models at any locations, moreover it gives fairly good predictions in the three locations. This is a clear advantage of the method, so it seems. Congratulations for this. Still future work should validate this model for many other locations in the globe to see the good performance is maintained.
L385 designed
Conclusions need to have a small sentence on the importance of this accurate method for practical purposes, including for example, coastal and offshore applications for marine energy (e.g. 10.1016/j.renene.2019.10.014), in marine engineering for coastal and offshore assets management and design (e.g. 10.1680/jmaen.2019.172.3.71 or 10.1680/jmaen.2020.173.4.96) and integrated coastal zones management (e.g. 10.5894/RGCI-N390). Note that these are just examples please you are encouraged to use others as well.
Author Response
We would like to thank you for your constructive comments concerning our article (Manuscript ID: jmse-1494298). These comments are all valuable and helpful for improving our article. All the authors have seriously discussed about all these comments. According to the reviewers’ comments, we have tried our best to modify our manuscript to satisfy with the requirements of this journal and readers. In this revised version, changes to our manuscript within the document were all highlighted by using red colored text.

Reviewer 2 Report
In the present paper the authors are presenting a new deep NN model for long term prediction of significat wave height. The paper is well structured and can be accepted for publication. Below are suggestions to the authors in order the authors to improve their paper:
-the authors should describe better the originality of the paper. What did you develop from scratch? If you just used already developed routines of matlab of any other tool the method is not novel but in generalthe framework is publishable. Please explain better the level of originality.
-Please state clealry that the proposed method is effective ONLY for the data set under study.
-More details of the overall frafework should be presented (eg what is input from one model and what is output to the other).
-More references should be added from published work in the past 5 years related to nowcastingand forecasting of ocean waves
-please explain what are the differences of long-term with rest prediction time periods.
Author Response

(The authors gave the same response as above.)
